# Regularity Normalization: Neuroscience-Inspired Unsupervised Attention across Neural Network Layers [note 1]

**DOI:** 10.3390/e24010059

**Published:** 2021-12-28

**Authors:** Baihan Lin

**Affiliations:** 1Department of Neuroscience, Columbia University Irving Medical Center, New York, NY 10032, USA; baihan.lin@columbia.edu; 2Department of Systems Biology, Columbia University Irving Medical Center, New York, NY 10032, USA

**Keywords:** neuronal coding, biologically plausible models, minimum description length, deep neural networks, normalization methods

## Abstract

Inspired by the adaptation phenomenon of neuronal firing, we propose the regularity normalization (RN) as an unsupervised attention mechanism (UAM) which computes the statistical regularity in the implicit space of neural networks under the Minimum Description Length (MDL) principle. Treating the neural network optimization process as a partially observable model selection problem, the regularity normalization constrains the implicit space by a normalization factor, the universal code length. We compute this universal code incrementally across neural network layers and demonstrate the flexibility to include data priors such as top-down attention and other oracle information. Empirically, our approach outperforms existing normalization methods in tackling limited, imbalanced and non-stationary input distribution in image classification, classic control, procedurally-generated reinforcement learning, generative modeling, handwriting generation and question answering tasks with various neural network architectures. Lastly, the unsupervised attention mechanisms is a useful probing tool for neural networks by tracking the dependency and critical learning stages across layers and recurrent time steps of deep networks.

## 1. Introduction

The Minimum Description Length (MDL) principle asserts that the best model given some data is the one that minimizes the combined cost of describing the model and describes the misfit between the model and data [1] with a goal to maximize regularity extraction for optimal data compression, prediction and communication [2]. Most unsupervised learning algorithms can be understood using the MDL principle [3], treating the neural net (NN) as a system communicating the input to a receiver.

If we consider the neural network training as the optimization process of a communication system, each input at each layer of the system can be described as a point in a low-dimensional continuous constraint space [4]. If we consider the activations from each layer of a neural network as the population codes, the constraint space can be subdivided into the input-vector space, the hidden-vector space, and the *implicit space*, which represents the underlying dimensions of variability in the other two spaces, i.e., a reduced representation of the constraint space. For instance, if we are given an image of an object, the rotated or scaled version of the same image still refers to the same objects, then each instance of the object can be represented by a code assigned position on a 2D implicit space with one dimension as orientation and the other as size of the shape [4]. The relevant information about the implicit space can be constrained to ensure a minimized description length of the networks.

In this paper, we adopt a similar definition of implicit space as in [4], but extend it beyond unsupervised learning, into a generic neural network optimization problem in both supervised and unsupervised settings [5]. In addition, we consider the formulation and computation of description length differently. Instead of considering neural networks as population codes, we formulate each layer of neural networks during training a state of model selection. In our setup, the description length is computed not in the scale of the entire neural networks, but by the unit of each layer of the network. In addition, the optimization objective is not to minimize the description length, but instead, to take into account the minimum description length as part of the normalization procedure to reparameterize the activation of each neurons in each layer. The computation of the description length (or model cost as in [4]) aims to minimize it, while we directly compute the minimum description length in each layer not to minimize anything, but to reassign the weights based on statistical regularities. Finally, we compute the description length by an optimal universal code obtained by the batch input distribution in an online incremental fashion. This model description serves both as a normalization factor to speed up training and as a useful lens to analyze the information propagation in deep networks. As this approach offers internal regularization across layers to emphasize the gradients of a subset of activating units, acting as a saliency filter over activations, we call this framework the *Unsupervised Attention Mechanism* (UAM, see Figure 1).

Our main contribution is two-fold: (1) From the engineering perspective, the work offers a performance improvement of numerical regularization in a imbalanced input data distribution; (2) More importantly, from the analytical perspective, we consider the proposed method a novel way of analyzing and understanding the deep networks during training, learning and failing, beyond its empirical advantages on the non-stationary task setting in the result section. The main point is not simply about beating the state-of-the-art normalization method with another normalization, but more to offer a new perspective where people can gain insights of the deep network in action—through the lens of model complexity characterized by this normalization factor the model computes along the way. On the subsidiary numerical advantage of the proposed method, the results suggest that combining regularity normalizations with traditional normalization methods can be much stronger than any method by itself, as their regularization priors are in different dimensions and subspaces.

## 2. Related Work

### 2.1. Neuroscience Inspirations

In the biological brains of primates, high-level brain areas are known to send top-down feedback connections to lower-level areas to encourage the selection of the most relevant information in the current input given the current task [6], similar to the communication system above. This type of modulation is performed by collecting statistical regularity in a hierarchical encoding process between these brain areas. One feature of the neural coding during the hierarchical processing is the adaptation: in vision neuroscience, vertical orientation reduce their firing rates to that orientation after adaptation [7], while the cell responses to other orientations may increase [8]. These behaviors contradict to the Bayesian assumption that the more probable the input, the larger firing rate should be, but instead, well match the information theoretical point-of-view that the most relevant information (the saliency), which depends on the statistical regularity, have higher “information”, just as the firing of the neurons. As [9] hypothesizes that the firing rate represents the code length instead of probability, similarly, the more regular the input features are (e.g., after adaption), the lower it should yield the activation, thus a shorter code length of the model (a neuron or a population).

One distinction should be noted between developing computational models to infer neurobiological functions and developing neuroscience-inspired algorithms for engineering problems. This study is an example of the latter. It is known that the working manner of the neuroscience in human brain is not unsupervised in most cases, especially in the events of top-down control and task-driven behaviors. For instance, supervised learning models, such as pretrained deep neural networks, are considered a candidate model in various behavioral tasks, such as recognizing images [10,11]. The evolutionary and learning processes of biological brains are also commonly studied in the reinforcement learning domain [12,13]. The inspiration we primarily draw from neuroscience to this work, is the bottom-up neuronal firing patterns driven by regularity. As a result, the proposed method consists of an unsupervised attention mechanism. Despite this distinction, in later section we will introduce the flexibility for the proposed methods to include supervised signals and feedbacks, as a data prior for saliency extraction and top-down attention.

### 2.2. Normalization Methods in Neural Networks

The batch normalization (BN) performs a global normalization along the batch dimension such that for each neuron in a layer, the activation over all the mini-batch training cases follows standard normal distribution, reducing the internal covariate shift [14]. Similarly, the layer normalization (LN) performs a global normalization over all the neurons in a layer, and have shown effective stabilizing effect in the hidden state dynamics in recurrent networks [15]. The weight normalization (WN) applies the normalization over the incoming weights, offering computational advantages for reinforcement learning and generative modeling [16]. Like batch normalization and layer normalization, we apply the normalization on the activation of the neurons, but as an element-wise reparameterization (over both the layer and batch dimension), which we term the regularity normalization (RN). In Section 5.2, we also propose several variants of our approach with batch-wise and layer-wise reparameterization, such as the regularity batch normalization (RBN) and the regularity layer normalization (RLN).

### 2.3. Description Length in Neural Networks

Ref. [17] first introduces the description length to quantify the simplicity of neural networks and develop an optimization method to minimize the amount of information required to communicate the weights of the neural network. Ref. [4] considers the neural networks as population codes and uses the MDL to develop highly redundant population code. They show that by assuming the hidden units reside in low-dimensional implicit spaces, optimization process can be applied to minimize the model cost under MDL principle. Our proposed method adopts a similar definition of implicit space, but considers the implicit space as the data-dependent encoding statistical regularities. Unlike [4,17], we consider the description length as a indicator of the data input and assume that the implicit space is constrained when we normalize the activation of each neurons given its statistical regularity. Ref. [18] computes the codelength of the entire network and showed that variational methods are ineffective to compute network complexity. Unlike [18], we approximate the codelength across each pairs of the neural network layers with an incremental formulation.

### 2.4. Attention Maps

One of the main question addressed in this paper is how to create an attention map to obtain an optimal and efficient model. Previous work have proposed useful methods like the self-attention [19] or pruning [20,21], where the attention maps are usually learned given an objective function. Our work is related to the residual attention network [22] which stacks attention modules to create an interaction between features and attention masks. Similar to this motivation, we instead propose to use layer-specific activation regularity to create residual attention. Likewise, these layer-specific attention maps change adaptively as the layers go deeper. Dual attention network is another related model architecture which computes attention on both the spatial and channel dimensions [23]. Similar to their work, the attention (in our case, the regularity) can be computed element-wise, batch-wise or layer-wise depending on the tasks. The attention can also be utilized in temporal dimension. For instance, ref. [24] and its followup work [25,26] introduce a spatiotemporal attention mechanism in recurrent neural networks that can dynamically refine the observed motion information in group activity recognition and skeleton joint modeling. In Section 6.2, we demonstrate the flexibility to also apply the regularity attention to the unfolded time steps in recurrent networks. Unlike these related works, in this work we compute the unsupervised attention maps by directly estimating the minimum description length given the network states without post hoc optimization from the loss, in a process driven only by the law of parsimony. Self-attention is also effectively studied in unsupervised regimes in natural language domain (e.g., masked language model). This unsupervised property distinguishes our method with most attention mechanisms in many other machine learning domains.

## 3. Background

### 3.1. Minimum Description Length

Given a model class Θ consisting of a finite number of models parameterized by the parameter set θ. Given a data sample *x*, each model in the model class describes a probability P(x|θ), as the probability of observing a data sample x given the model parameter θ, with its code length computed as −logP(x|θ). The minimum code length given any arbitrary θ would be given by $L\left(x|\hat{\theta }\left(x\right)\right)=-\log P\left(x|\hat{\theta }\left(x\right)\right)$ with model $\hat{\theta }\left(x\right)$ which compresses data sample *x* most efficiently and offers maximum likelihood $P(x|\hat{\theta (x)})$ [2]. However, the compressibility of the model, computed as the minimum code length, can be unattainable for multiple non-i.i.d. data samples as individual inputs, as the probability distributions of most efficiently representing a certain data sample *x* given a certain model class can vary from sample to sample. The solution relies on the existence of a universal code, $\bar{P}\left(x\right)$ defined for a model class Θ, such that for any data sample *x*, the shortest code for *x* is always $L(x|\hat{\theta }\left(x\right))$, as shown in [27].

### 3.2. Normalized Maximum Likelihood

The minimum description length problem is often addressed by trying to find the model that is “closet” to the true distribution f(·) in some well-defined sense. To select for a proper optimal universal code, a cautious approach would be to assume a worst-case scenario in order to make “safe” inferences about the unknown world. Formally, the worst-case expected regret is given by $R\left(p∥\Theta \right)=\max_{q}E_{q}\left[\ln\frac{f(x|{\hat{\theta }}_{x})}{p(x)}\right]$, where p(·) and the “worst” distribution q(·) is allowed to be any probability distribution [28]. Without referencing the unknown truth, [27] formulates finding the optimal universal distribution as a mini-max problem of computing $p*=\arg\min_{p}\max_{q}E_{q}\left[\ln\frac{f(x|{\hat{\theta }}_{x})}{p(x)}\right]$, the coding scheme that minimizes the worst-case expected regret. Among the universal codes, the normalized maximum likelihood (NML) probability minimizes the worst-case regret and avoids assigning an arbitrary distribution to Θ. The minimax optimal solution is given by [29]:(1)$$\begin{array}{c} P_{NML}\left(x\right)=\frac{P(x|\hat{\theta }\left(x\right))}{\sum_{x^{\prime }}P\left(x^{\prime }|\hat{\theta }\left(x^{\prime }\right)\right)} \end{array}$$
where the summation is over the entire data sample space (i.e., all possible data sample, hence $x^{\prime }$). Figure 2 describes the optimization problem of finding optimal model $P(x_{i}|\hat{\theta_{i}})$ given the data sample $x_{i}$ among model class Θ. The models in the class, P(x|θ), are parameterized by the parameter set θ. $x_{i}$ are data sample from data *X*. With this distribution, the regret is the same for all data sample *x* given by [2]:(2)$$\begin{array}{c} \begin{array}{cc} COMP(\Theta ) & \equiv regret_{NML}\\ & \equiv -\log P_{NML}\left(x\right)+\log P\left(x|\hat{\theta }\left(x\right)\right)\\ & =\log\sum_{x^{\prime }}P\left(x^{\prime }|\hat{\theta }\left(x^{\prime }\right)\right) \end{array} \end{array}$$
which computes the log of the denominator in Equation (Equation 1) in its expanded form. The COMP defines the model class complexity, as it indicates how many different data samples can be well explained by the model class Θ.

## 4. Neural Networks as Model Selection

In the neural network setting where the optimization process is performed in batches (as incremental data sample $x_{i}$ with *i* denoting the batch *i*), the model selection process is formulated as a partially observable problem (as in Figure 3 and [5]). The generative model $P(x_{i}^{k}|\hat{\theta_{i}^{k}})$ is the function parameterized by $\hat{\theta_{i}^{k}}$ that maps from $x_{i}^{k-1}$ (input layer k−1) to $x_{i}^{k}$ (the activations of layer *k*) at training time *i*. Herein to illustrate our approach, we consider a feedforward neural network as an example, without loss of generalizability to other model architectures (such as convolutional layers or recurrent modules). $x_{i}^{k}$ refers to the activation at layer *k* at time point *i* (the batch *i*). $\hat{\theta_{i}^{k}}$ is the parameters that describes $x_{i}^{k}$ (i.e., weights for layer k−1) optimized after i−1 steps (seen batches 0 through i−1). Because one cannot exhaust the search among all possible θ, by definition of the optimization process via gradient descent, we assume that the optimized parameter $\hat{\theta_{i}^{k}}$ at time step *i* (having seen batch 0 through i−1) is the optimal model $P(x_{i}^{k}|\hat{\theta_{i}^{k}})$ for the seen data sample $x_{i}^{k}$. Take layer *k* as our target of study, in later notations we will drop the superscript *k* to simply the formulation, without loss of generality to other layers or computing modules. To continue, we generalize the optimal universal code with the normalized maximum likelihood (NML) formulation:(3)$$\begin{array}{c} P_{NML}\left(x_{i}\right)=\frac{P(x_{i}|{\hat{\theta }}_{i}\left(x_{i}\right))}{\sum_{j=0}^{i}P\left(x_{j}|{\hat{\theta }}_{j}\left(x_{j}\right)\right)} \end{array}$$
where ${\hat{\theta }}_{i}\left(x_{i}\right)$ refers to the model parameter already optimized for i−1 steps and have seen sequential data sample $x_{0}$ through $x_{i-1}$. This distribution is updated every time a new data sample is fed, and can thus be computed incrementally in batches.

*Choice of optimization methods:* The proposed framework is agnostic to the optimization methods and feedback signals due to its unsupervised nature. Therefore, most neural network training methods should be by default compatible to this model selection setting and the undermentioned normalization methods within the framework. For the empirical evaluations in the following sections, we use the stochastic gradient descent as neural net optimization method by convention.

*Disclaimer and theoretical gaps:* In traditional model selection problems, the MDL can be regarded as ensemble modeling process and usually involves multiple models. However, in our neural network problem, we assume that the only model trained at each step is the local “best” model learned so far, i.e., a partially observable model selection problem. This implies that the local maximal likelihood may not be a global best solution for model combinations, because the generation of optimized parameter set for a specific layer currently adopts greedy approach, such that the model selection could be optimized for each step. This assumption loosely links the computed *regularity* with, (if not eventually converges to), the *MDL* in theory. We don’t claim the computed universal code in our framework is an exact MDL metric, but instead, only provide it as a useful measure of the model complexity and code length of the network layers. Further theoretical work is worth pursuing to demonstrate whether this greedy approach converges to the best global selection.

## 5. The Unsupervised Attention Mechanism

### 5.1. Standard Case: Regularity Normalization

We first introduce the standard formulation of the unsupervised attention mechanism, the regularity normalization (RN). As outlined in Algorithm 1, the input is the activation of each neurons in certain layer and batch. Parameters COMP and θ are updated after each batch, through the incrementation in the normalization and optimization in the training respectively. As the numerator of $P_{NML}$ at this optimization step *i* of the normalization, the term $P(x_{i}|\hat{\theta_{t}}\left(x_{i}\right))$ is computed to be stored as a log probability of observing sample $x_{i}$ in $N(\mu_{i-1},\sigma_{i-1})$, the normal distribution with the mean and standard deviation of all past data sample history ($x_{0},x_{1},\cdots ,x_{i-1}$), with a prior for $P(x|\hat{\theta }\left(x\right))$. For numerical evaluations, here we select a Gaussian prior based on the assumption that each *x* is randomly sampled from a Gaussian distribution, and the parameter sets from model class Θ are Gaussian before nonlinear activation functions, but the framework can theoretically work with any arbitrary prior. We discuss this Gaussian assumption and provided an extension with other priors (see Appendix A for a possible extension).

As in Equation (Equation 2), COMP is the denominator of $P_{NML}$ taken log, so the “*increment*” function takes in $COMP_{t}$ storing $\sum_{j=0}^{i-1}P\left(x_{j}|\hat{\theta_{t}}\left(x_{j}\right)\right)$ and the latest batch of $P(x_{i}|\hat{\theta_{t}}\left(x_{i}\right))$ to be added in the denominator, stored as $COMP_{t+1}$. The “*increment*” step involves computing the log sum of two values, which can be numerically stabilized with the log-sum-exp (LSE, also known as RealSoftMax) numerical trick conventionally used in machine learning optimization [30]. In continuous data streams or time series analysis, the incrementation step can be replaced by integrating over the seen territory of the probability distribution X of the data. The normalization factor is then computed as the shortest code length *L* given the normalized maximum likelihood, the universal code distribution in Equation (Equation 3).

Figure 4 is a flowchart outlining the training and testing pipelines with the regularity-based methods installed. In the training phase, the data are fed in batches, and after each layer is activated, the activations are reparameterized by the normalizations with Algorithm 1 before feeding into the next layer. Since the regularity computation has no trainable parameters, the optimization (e.g., stochastic gradient descent) acts on all neural net parameters except for the ones in the regularity modules. As a result, the regularity-based modules doesn’t require pretraining or optimization. In the testing phase, the regularity modules are turned off.
**Algorithm 1** Regularity Normalization (RN)**Input**: *x* over a mini-batch: $B=\{x_{1},\cdots ,x_{m}\}$;**Parameter**: $COMP_{t}$, $\hat{\theta_{t}}$**Output**: $y_{i}=RN\left(x_{i}\right)$$COMP_{t+1}$ = log(exp($COMP_{t}$) + $P(x_{i}|\hat{\theta_{t}}\left(x_{i}\right))$)$L_{x_{i}}=COMP_{t+1}-\log P\left(x_{i}|\hat{\theta_{t}}\left(x_{i}\right)\right)$$y_{i}=L_{x_{i}}*x_{i}$

### 5.2. Variant: Saliency Normalization

The normalized maximum likelihood distribution can be modified to also include a data prior function, s(x), given by [31] as $P_{NML}\left(x\right)=\frac{s\left(x\right)P(x|\hat{\theta }\left(x\right))}{\sum_{x^{\prime }}s\left(x^{\prime }\right)P\left(x^{\prime }|\hat{\theta }\left(x^{\prime }\right)\right)}$, where the data prior function s(x) can be anything, ranging from the emphasis of certain inputs, to the cost of certain data, or even the top-down attention. For instance, we can introduce the prior knowledge of the fraction of labels (say, in an imbalanced data problem where the oracle informs the model of the distribution of each label in the training phase); or in a scenario where we wish the model to focus specifically on certain feature of the input, say certain texture or color (just like a convolution filter); or in the case where the definition of the regularity drifts (such as the user preferences over years): in all these possible applications, the normalization procedure can be more strategic given these additional information. We formulate this additional functionality into regularity normalization, to be the saliency normalization (SN), where $P_{NML}$ is incorporated with a pre-specified data prior s(x). As illustrated in the orange track of Figure 4, when data priors are available, the top-down attention can serve as a useful input into the regularity modules for the saliency normalization.

### 5.3. Variant: Beyond Elementwise Normalization

In our current setup, the normalization is computed elementwise, considering the implicit space of the model parameters to be one-dimensional (i.e., all activations across the batch and layer are considered to be represented by the same implicit space). Instead, the definition of the implicit can be more than one-dimensional to increase the expressibility of the method, and can also be user-defined. For instance, we can also perform the normalization over the dimension of the batch, such that each neuron in the layer should have an implicit space to compute the universal code. We term this variant the regularity batch normalization (RBN). Similarly, we can perform regularity normalization over the layer dimension, as the regularity layer normalization (RLN). These two variants have the potential to inherit the innate advantages of the batch normalization and the layer normalization.

## 6. Demonstration: The Unsupervised Attention Mechanism as a Probing Tool

In this section, we provide a tutorial of how the unsupervised attention mechanism can be applied to understand how neural networks route relevant information during the learning process. We train two types of deep networks on the simple image classification problem with MNIST. In both experiments, training, validation and testing sets are shuffled into 55,000, 5000, and 10,000 cases. Batch size is set to 128. For optimization, stochastic gradient decent is used with learning rate 0.01 and momentum set to be 0.9. In both cases, we train the task to mastery (over 97%) for the vanilla feedforward and recurrent neural networks over 10 epochs. We record the change of the approximated minimum description length COMP (computed incrementally from the optimal universal code in Equation (Equation 3)) of each layer over the entire training time (time stamped by batches).

### 6.1. Over Different Layers in the Feedforward Neural Networks (FFNN)

In this analysis, we consider the classical 784-1000-1000-10 feedforward neural network, two hidden layers with ReLU activation functions. We compute the model code length of each layer of the network ($fc_{1}$ and $fc_{2}$) with respect to the last layer’s input. Figure 5A–C demonstrate the change of model complexity over training time. We observe that the model complexities increases smoothly and then gradually converges to a plateau, matching the information bottleneck hypothesis of deep networks. The COMP for the later (or higher) layer seems to be having a higher model complexity in the start, but after around 600 iterations, the earlier (or lower) layer seems to catch up in the model complexities. Comparing COMP curves of the neural nets with or without the proposed regularity normalization, we observe that the regularization appeared to implicitly impose a constraint to yield a lower model complexity in the earlier layers than the later layers. These behaviors are in need of further theoretical understanding, but in the next section, our empirical results suggest a benefit of this regularization on the performance across a variety of tasks.

### 6.2. Over Recurrent Time Steps in the Recurrent Neural Networks (RNN)

In this analysis, we consider a vanilla recurrent neural networks with 100 hidden units over 5 time steps with *tanh* activation functions. We computed the minimum description length of each time-unfolded layer of the recurrent network ($r_{1}$ to $r_{5}$) with respect to the last time-unfolded layer’s input. Similarly, we compare the COMP of each unfolded layers with respect to their input in the two networks, one with the regularity normalization installed at each time step (or unfolded layer), and one without the regularity normalization, i.e., the vanilla network. Since an unfolded recurrent network can be considered equivalent to a deep feedforward network, in this analysis, we wish to understand the effect of regularity normalization on the recurrent network’s layer-specific model complexity beyond the effect on a simply deeper feedforward neural net. We consider a recurrent unit to be also adapting to the statistical regularity not only over the training time dimension, but also the recurrent unfolded temporal dimension. In another word, we consider the recurrent units at different recurrent time step to be the same computing module at a different state in the model selection process (i.e., the same layer in Figure 3). Therefore, instead of keeping track of individual histories of the activations at each step, we only record one set of history for the entire recurrent history (by pooling all the activations from 0 to the current recurrent time steps).

As in Figure 5C,D, we observe that both networks adopts a similar model complexity progression. Unlike the traditional understanding of the asynchronous learning stages for different recurrent time steps, this analysis suggests that the change of the model complexity COMP during learning over different recurrent time steps are relatively universal. The model complexity of earlier time step (or earlier unfolded layer) seems to be much lower than later ones, and the margins are decreasing over the recurrent time step. This recurrent neural net appears to process the recurrent input in gradually increasing complexity until a plateau. As expected, the proposed regularity normalization regularizes the complexity assignments across recurrent time steps (as unfolded layers) such that the complexities of the later time steps remain a relatively low level. Further analysis on the recurrent networks with different architectures, multiple layers and different activations functions can enlighten more insights on these behaviors. In the next section, we included an example where regularity normalization improves the performance in recurrent generative modeling.

## 7. Empirical Results

### 7.1. The Imbalanced MNIST Problem with Feedforward Neural Network

As a proof of concept, we evaluate our approach on the MNIST task and compute the classification errors as a performance metric. As we specifically wish to understand the behavior where the data inputs are non-stationary and highly imbalanced, we create an imbalanced MNIST benchmark to test seven methods: batch normalization (BN), layer normalization (LN), weight normalization (WN), regularity normalization (RN) and its three variants: the saliency normalization (SN) with data prior as class distribution, regularity layer normalization (RLN) where the implicit space is defined to be layer-specific, and LN+RN which is a combined approach where the regularity normalization is applied after the layer normalization. Given the nature of the regularity normalization, it should better adapt to the regularity of data distribution than others, tackling the imbalanced problem by up-weighting the activation of the rare features and down-weighting dominant ones.

*Experimental setting.* The imbalanced degree *n* is defined as following: when n=0, it means that no classes are downweighted, so we term it the *“fully balanced”* scenario; when n=1 to 3, it means that a few cases are extremely rare, so we term it the *“rare minority”* scenario. When n=4 to 8, it means that the multi-class distribution are very different, so we term it the *“highly imbalanced”* scenario; when n=9, it means that there is one or two dominant classes that is 100 times more prevalent than the other classes, so we term it the *“dominant oligarchy”* scenario. In real life, *rare minority* and *highly imbalanced* scenarios are very common, such as predicting the clinical outcomes of a patient when the therapeutic prognosis data are mostly tested on one gender versus the others. More details of the setting can be found in the Section B.1.

*Performance.*Table 1 reports the test errors (in %) with their standard errors of the eight methods in 10 training conditions over two heavy-tailed scenarios: labels with under-represented and over-represented minorities. In the balanced scenario, the proposed regularity-based method doesn’t show clear advantages over existing methods, but still manages to perform the classification tasks without major deficits. In both the “rare minority” and “highly imbalanced” scenarios, regularity-based methods performs the best in all groups, suggesting that the proposed method successfully constrained the model to allocate learning resources to the “special cases” which are rare and out of normal range, while the BN and WN fail to learn it completely (ref. confusion matrices in the Section B.2). We observe that, In certain cases, like n = 0, 8 and 9 (“balanced” or “dominant oligarchy”), some of the baselines are doing better than RN. However, that is expected because those are the three cases in the ten scenarios that the dataset is more “regular” (or homogenous in regularity), and the benefit from normalizing against the regularity is expected to be minimal. For instance, in the “dominant oligarchy” scenario, LN performs the best, dwarfing all other normalization methods, likely due to its invariance to data rescaling but not to recentering. However, as in the case of n=8, LN + RN performs considerably well, with performance within error bounds to that of LN, beating other normalization methods by over 30%. On the other hand, if we look at the test accuracy results in the other seven scenarios, especially in the highly imbalanced scenarios (RN variants over BN/WN/baseline for around 20%), they should provide promises in the proposed approach ability to learn from the extreme regularities.

We observe that LN also manages to capture the features of rare classes reasonably well in other imbalanced scenarios, comparing to BN, WN and baseline. The hybrid methods RLN and LN + RN both display excellent performance in imbalanced scenarios, suggesting that combining regularity-based normalization with other methods is advantageous, as their imposed priors are in different dimensions and subspaces. This also brings up an interesting concept in applied statistics, the “no free lunch” theorem. LN, BN, WN and the proposed regularity-based normalizations are all developed under different inductive bias that the researchers impose, which should yield different performance in different scenarios. In our case, the assumption that we make, when proposing the regularity-based approach, is that in real life, the distribution of the encounter of the data samples are not necessarily i.i.d, or uniformly distributed, but instead follows irregularities unknown to the learner. The data distribution can be imbalanced in certain learning episodes, and the reward feedback can be sparse and irregular. This inductive bias motivates the development of this regularity-based method.

The results presented here mainly deal with the short term domain to demonstrate how regularity can significantly speed up the training. The long term behaviors tend to converge to a balanced scenario since the regularities of the features and activations will become higher (not rare anymore), with the normalization factors converging to a constant in a relative sense across the neurons. As a side note, despite the significant margin of the empirical advantages, we observe that regularity-based approach offers a larger standard deviation in the performance than the baselines. We suspect the following reason: the short-term imbalanceness should cause the normalization factors to be considerably distinct across runs and batches, with the rare signals amplified and common signals tuned down; thus, we expect that the predictions to be more extreme (the wrong more wrong, the right more right), i.e., a more stochastic performance by individual, but a better performance by average. We observe this effect to be smaller in the long term (when the normalization factors converge to a constant relatively across neurons). Future further analysis might help us fully understand these behaviors in different time scales (e.g., the converging performance over 100 epochs).

*Probing the network with the unsupervised attention mechanism*. Figure 6A–C demonstrate three steoretypical COMP curves in the three imbalanced scenarios. In all three cases, the later (or higher) layer of the feedforward neural net adopts a higher model complexity. The additional regularity normalization seems to drive the later layer to accommodate for the additional model complexities due to the imbalanced nature of the dataset, and at the same time, constraining the low-level representations (earlier layer) to have a smaller description length in the implicit space. This behaviors matches our hypothesis how this type of regularity-based normalization can extract more relevant information in the earlier layers as inputs to the later ones, such that later layers can accommodate a higher complexity for subsequent tasks. Figure 6D,E compare the effect of imbalanceness on the COMP difference of the two layers in the neural nets with or without RN. We observe that when the imbalanceness is higher (i.e., *n* is smaller), the neural net tends to maintains a significant complexity difference for more iterations before converging to a similar levels of model complexities between layers. On the other hand, the regularity normalization constrains the imbalanceness effect with a more consistent level of complexity difference, suggesting a possible link of this stable complexity difference with a robust performance in the imbalanced scenarios.

### 7.2. The Classic Control Problem in OpenAI Gym with Deep Q Networks (DQN)

We further evaluate the proposed approach in the reinforcement learning problem, where the rewards can be sparse. For simplicity, we consider the classical deep Q network (DQN) [32] and test it in the OpenAI Gym’s LunarLander and CarPole environments [33] (see Section C.1 for the neural network setting and game setting). We evaluate five agents (DQN, +LN, +RN, +RLN, +RN+LN) by the final scores in 2000 episodes across 50 runs.

*Performance.* As in Figure 7A, in the LunarLander environment, DQN + RN (66.27 ± 2.45) performs the best among all five agents, followed by DQN + RN + LN (58.90 ± 1.10) and DQN+RLN (51.50 ± 6.58). All three proposed agents beat DQN (43.44 ± 1.33) and DQN-LN (51.49 ± 0.57) by a large marginal. The learning curve also suggests that taking regularity into account speed up the learning and helped the agents to converge much faster than the baseline. Similarly in the CarPole environment, DQN + RN + LN (206.99 ± 10.04) performs the best among all five agents, followed by DQN + RN (193.12 ± 14.05), beating DQN (162.77 ± 13.78) and DQN + LN (159.08 ± 8.40) by a large marginal (Figure 7D). These numerical results suggests the proposed method has the potential to benefit the neural network training in reinforcement learning setting. On the other hand, certain aspects of these behaviors are worth further exploring. For example, the proposed methods with highest final scores do not converge as fast as DQN + LN, suggesting that regularity normalization resembles adaptive learning rates which gradually tune down the learning as scenario converges to stationarity.

*Probing the network with the unsupervised attention mechanism*. As shown in Figure 7B,C, the DQN has a similar COMP change during the learning process to the ones in the computer vision task. We observe that the COMP curves are more separable when regularity normalization is installed. This suggests that the additional regularity normalization constrains the earlier (or lower) layers to have a lower complexity in the implicit space and saving more complexities into higher layer. The DQN without regularity normalization, on the other hand, has a more similar layer 1 and 2. We also observe that the model complexities COMP of the target networks seems to be more diverged in DQN comparing with DQN+RN, suggesting the regularity normalization as a advantageous regularization for the convergence of the local to the target networks. Figure 7E,F offer some intuition why this is the case: the DQN+RN (red curve) seems to maintain the most stable complexity difference between layers, which stabilizes the learning and provides a empirically advantages trade-off among the expressiveness of layers given a task (see Section C.3 for a complete spectrum of this analysis).

### 7.3. The Generative Modeling Problem with Deep Recurrent Attentive Writer (DRAW)

We investigate the effect of the regularity normalization on the generative modeling of variational models. We evaluate the generative modeling task on MNIST dataset with the state-of-the-art model, the Deep Recurrent Attention Writer (DRAW) [34]. Following the experimental setup as [15], we record the KL divergence variational bound for the first 50 epochs. Figure 7G highlights the speedup benefit of applying the regularity normalization even faster than the layer normalization. Other than the regularity normalization and layer normalization, we also note a significant speedup over baseline in the previously underperforming saliency normalization. In the classification task, we don’t spend time choosing the best data prior for the saliency normalization, but this prior of class information happens to be beneficial to this generative task. We would like to comment that the potential to integrate top-down priors in a neural network can be impactful, especially in the generative models.

### 7.4. The Dialogue Modeling Problem in bAbI with Gated Graph Neural Networks (GGNN)

We further test whether we can impose the unsupervised attention mechanism beyond the layers of neural nets, into the message passing across generic neural modules. We choose to evaluate on a natural language processing dialogue modeling task, the bAbI [35] benchmark, a dataset for text understanding and reasoning. We apply different normalizations upon its state-of-the-art, the Gated Graph Sequence Neural Networks (GGNN) [36] (experimental details in Appendix D). As the bAbI benchmark is a solved task (with 100% accuracy in all the GGNN variants), so we focus on the short-term learning sessions within the first few epochs. As show in Figure 7H,I, we observe that in the two most difficult tasks (task 18 and task 19), the GGNN + RN + LN converges the fastest among all agents, followed by the layer normalization. Although the benefit of the regularity-based normalization is not apparrent in this task, this experiment shows that the unsupervised attention mechanism can be potentially useful for more complicated information propagations among computational modules, as in this graph neural network example.

### 7.5. Procedurally-Generated Reinforcement Learning in MiniGrid with Proximal Policy Optimization (PPO)

Lastly, we experiment on whether the regularity-based priors are useful in more complicated reinforcement learning environments. In MiniGrid [37], environments are procedurally-generated with different regularities across states. These states usually involve multi-modal signals like text-based commands and pixel-based image states. We introduce the normalization methods to a popular policy gradient algorithm, Proximal Policy Optimization (PPO) [38], and demonstrate that PPO + RN + LN can be very robust. For instance, in the RedBlueDoors, the most challenging memory-based task, while all the agents converge in the later rounds, we observe that in the earliest frames, our methods receive more rewards by exploring based on regularity (Figure 7J). Due to the unstable nature of this open-sourced benchmark in development, this experiment is not conclusive on a constant performance improvement of the proposed regularity-based normalizations. The fast convergence of regularity-boosted agent in early rounds might suggest a benefit of exploring unseen states modulated by the regularity of the learned representations in policy gradient algorithms.

## 8. Discussion

Inspired by the neural code adaptation of biological brains, in this paper we introduce a biologically plausible unsupervised attention mechanism taking into account the regularity of the activation distribution in the implicit space, and normalizing it to upweight activation for rarely seen scenario and downweight activation for commonly seen ones. This work provides an example on how neuronal phenomena can offer straightforward solutions to important engineering questions such as when, where and how to allocate resources most productively in the learning process.

This method can complement existing measures to understand deep learning. Despite the success of deep learning systems in various engineering applications, the interpretability of its results and the comprehensive theoretical understanding of its effectiveness are still limited. This prevents confident deployments on critical ares like medical diagnosis and credit analysis. In spirit of the *no free lunch* theorem, the existing measures of the deep network each offer valuable but limiting insights from different perspectives. Our estimation of the minimum description length is related to other complexity and information-theoretical measures, but differs in several critical ways:

*Mutual information (MI)*: In the traditional perspective, the minimum description length is a two-part code, the sum of the data code length and the model code length: L=L(x|M)+L(M). The mutual information only accounts for the data code length. [39] analyzes the mutual information that each layer preserves on the input and output variables and suggested that the goal of the network is to optimize the Information Bottleneck (IB) tradeoff between compression and prediction, successively, for each layer. Despite various recent attempts to estimate the mutual information [40,41], the information bottleneck approach can arrive at different conclusions under different assumptions for these estimation and misleading causal connections between compression and generalization [42]. Our MDL estimate complements this missing link by (1) incorporating also the model code length and (2) computing the normalized maximum likelihood (NML) incrementally with simple inference techniques. Unlike many mutual information estimators, our NML-based MDL estimate has no arbitrary parameters like the number of bins. While a high mutual information (or L(x|M)) is necessary for effective compression, a low model complexity (or L(M)) can further ensure a parsimonious representation. Fundamentally, we implement it with an incremental update of normalized maximum likelihood, constraining the implicit space to have a low model complexity and short universal code length.

*VC dimension [43] and Rademacher complexity [44]*: Both the Rademacher complexity and VC dimension depend only on the data distribution and model architecture, and not on the training procedure used to find models. Our proposed estimate of model complexity, on the other hand, depends on the training procedure and captures the dynamics over training. This additional sensitivity to the training procedures and training stages can offer us critical information to understand the deep learning’s fitting behavior and monitor the states of the deep learning models in real time.

*Effective model complexity (EMC) [45]*: The effective model complexity is the maximum number of samples on which it can achieve close to zero training error, and is proposed to specifically analyze the “double-descent” phenomenon in deep learning [46]. Unlike EWC, our model complexity estimate COMP is a function over time, instead of just one number as the EWC. This property allows us to track the COMP trajectories over different training states. Unlike EWC, COMP does not have to depend on the true labels of the data distribution and can therefore apply to more scenarios beyond the double descent, such as designing meta-learning algorithms (to adapt based on complexity) and selecting the right neural network at the right moment based on MDL.

In theory, our study suggests that the new regularization mechanism offers a dynamic view of the model complexity during training. In machine learning community, model complexity is related to a model’s generalization ability. Despite not a focus of this work, it is our next step to understand the link between between our proposed approximation form of minimum description length and generalization in the deep networks (e.g., via PAC-learning bound). Other than the empirical and theoretical insights to the deep learning community, there are potential broader impacts to other active research domains from our study as well:

*Promote AI fairness and diversity:* Recent AI systems and applications have been extensively deployed not only in daily life, but also in many sensitive environments to make important and life-changing decisions such as forensics, healthcare, credit analysis and self-driving vehicles. It is crucial to ensure that the decisions do not reflect discriminatory behavior toward certain groups or populations. The most common bias is the representation bias in existing datasets such as underrepresenting African and Asian communities in 23andMe genotype dataset [47] and lacking geographical diversity in ImageNet dataset [48]. There are also unintentional types of bias. For instance, [49] showed that vanilla PCA on the labeled faces in the wild (LFW) dataset has a lower reconstruction error rate for men than for women faces, even if the sampling is done with an equal weight for both genders. In face of these challenges, our proposed methods implicitly incorporate the fairness by allocating more learning resources to underrepresented samples and features through the normalization process. In addition, we propose the saliency normalization, which can introduce top-down attention and data prior to facilitate representation learning given fairness criteria.

*Incorporate useful data priors for security:* Other than fairness, the proposed normalization allows to incorporate top-down priors of other kinds. For instance, security-related decision making systems usually requires stability in certain modules and adaptability in others. Feeding the learning states as data prior into the proposed normalization can individually constrain each modules to accomplish this goal. Given existing knowledge, one can also flag certain feature types as important (or “salient”) in anomaly detection or information retrieval tasks to allocate more learning resources. Integration of top-down and bottom information is crucial for developing neural networks which can incorporate priors. One main next direction of this research is the top-down attention given by data prior (such as feature extracted from signal processing, or task-dependent information). The application of top-down attention s(x) to modulate the normalization process can vary in different scenarios. Further investigation of how different functions of s(x) behave in different task settings may complete the story of having this method as a top-down meta learning algorithm potentially advantageous for continual multitask learning. Imposed on the input data and layer-specific activations, unsupervised attention mechanism has the flexibility to directly install top-down attention from either oracle supervision or other meta information.

*Draw attention to neuroscience-inspired algorithms:* Like deep learning, many fields in artificial intelligence (AI) benefited from a rich source of inspirations from the neuroscience for architectures and algorithms. While the flow of inspirations from neuroscience to machine learning has been sporadic [50], it systematically narrows the major gaps between humans and machines: the size of required training datasets [51], out-of-set generalization [52], adversarial robustness [53,54], reinforcement learning [13,55,56] and model complexity [57]. Biological computations which are critical to cognitive functions are usually excellent candidates for incorporation into artificial systems and the neuroscience studies can provide validation of existing AI techniques for its plausibility as an integral component of an overall general intelligence system [58].

## 9. Conclusions

In summary, inspired by the biological phenomenon of the regularity-driven neuronal firing, we propose to consider the neural network training process as a model selection problem and compute the model complexity of a neural network layer as the optimal universal code length with a normalized maximum likelihood formulation. We show that this code length can serve as a normalization factor and can be easily incorporated with established regularization methods to (1) speed up training, (2) increase the sensitivity to imbalanced data or feature spaces, and (3) analyze and understand neural networks in action via the lens of model complexity.

## Figures and Tables

**Figure 1 entropy-24-00059-f001:**
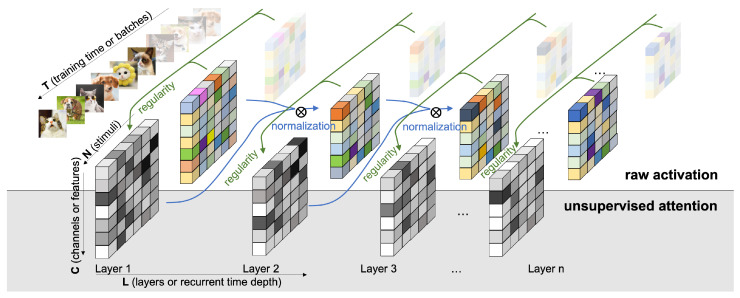
**Unsupervised attention mechanism** computes and normalizes regularity sequentially across layers.

**Figure 2 entropy-24-00059-f002:**
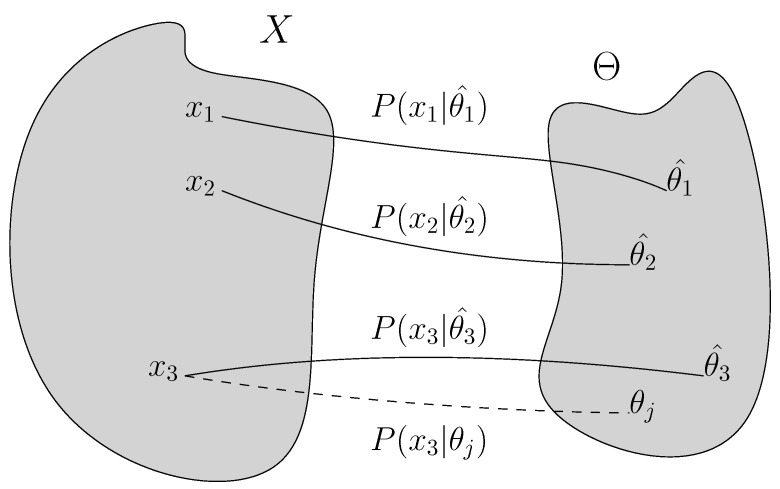
**Normalized maximal likelihood.** In this illustration, data sample $x_{i}$ are drawn from the entire data distribution *X* and model $\hat{\theta_{i}}$ is the optimal model that describes data $x_{i}$ with the shortest code length. $\theta_{j}$ is an arbitrary model that is not $\hat{\theta_{3}}$, so $P(x_{3}|\theta_{j})$ is not considered when computing optimal universal code according to Equation (Equation 1).

**Figure 3 entropy-24-00059-f003:**
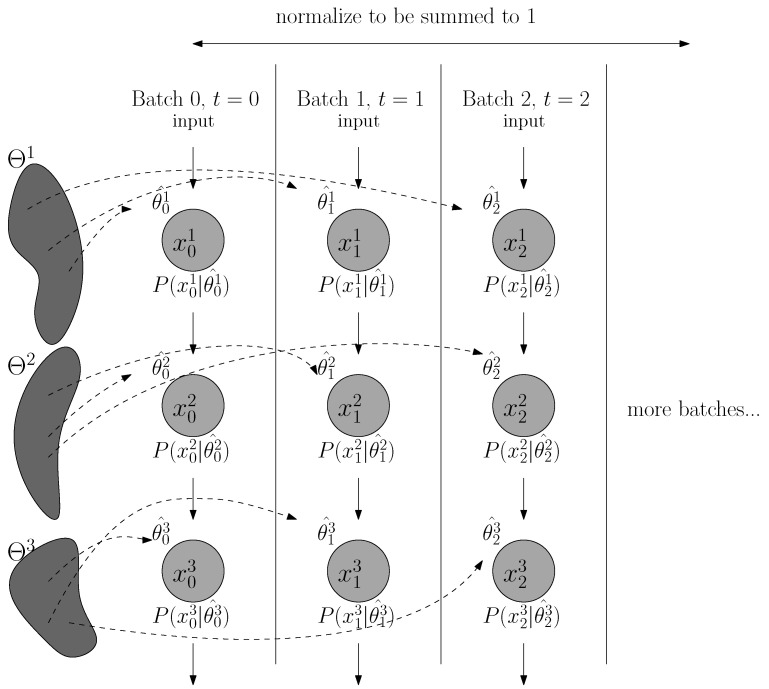
**Model selection process in neural network.** If we consider each time step of the optimization (drawn here to be batch-dependent) as the process of choose the optimal model from model class $\Theta^{k}$ for the *k*th layer (or the *k*th computing module) of the neural networks, the optimized parameter $\hat{\theta_{i}^{k}}$ with subscript *i* as time step t=i and superscript *k* as layer *k* can be assumed to be the optimal model among all models in the model class $\Theta^{k}$. The normalized maximum likelihood can be computed by choosing $P(x_{i}^{k}|\hat{\theta_{i}^{k}})$, the “optimal” model with shortest code length given data $x_{i}^{k}$, as the summing component in normalization.

**Figure 4 entropy-24-00059-f004:**
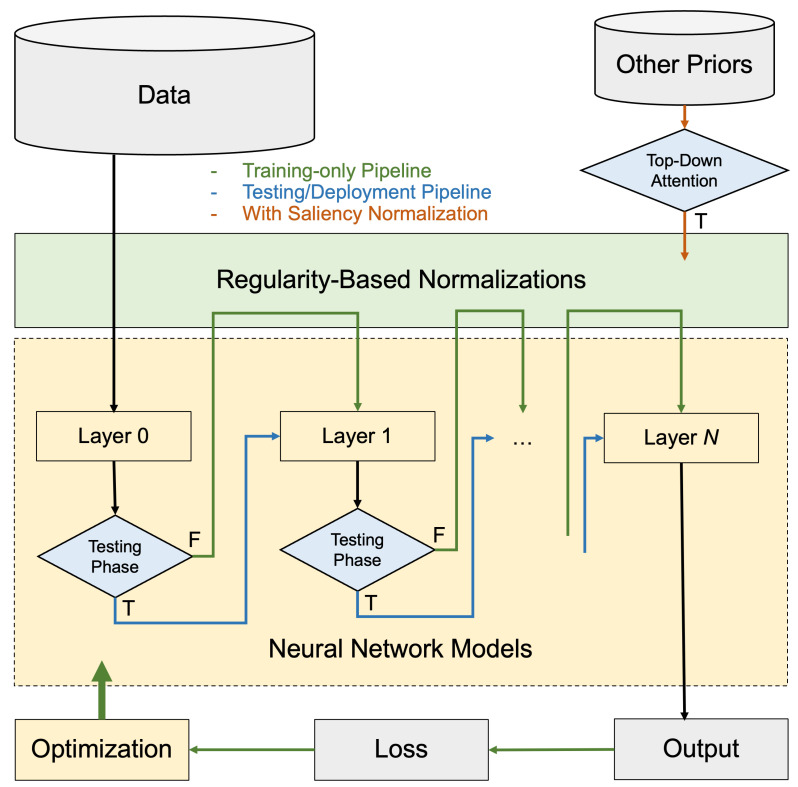
**Training and testing pipelines with regularity-based normalizations**: The flowchart outlines the three possible tracks with the regularity-based methods installed. In the training phase, the data is fed in batches, and after each layer is activated, the activations are reparameterized by the normalizations before feeding into the next layer. The optimization (e.g., stochastic gradient descent) acts on all neural net parameters except for the ones in the regularity modules. The regularity-based modules doesn’t require pretraining or optimization. In the testing phase, the regularity modules are turned off. When certain priors are available, the top-down attention can serve as a useful input into the regularity modules, as illustrated in the orange track for the saliency normalization.

**Figure 5 entropy-24-00059-f005:**
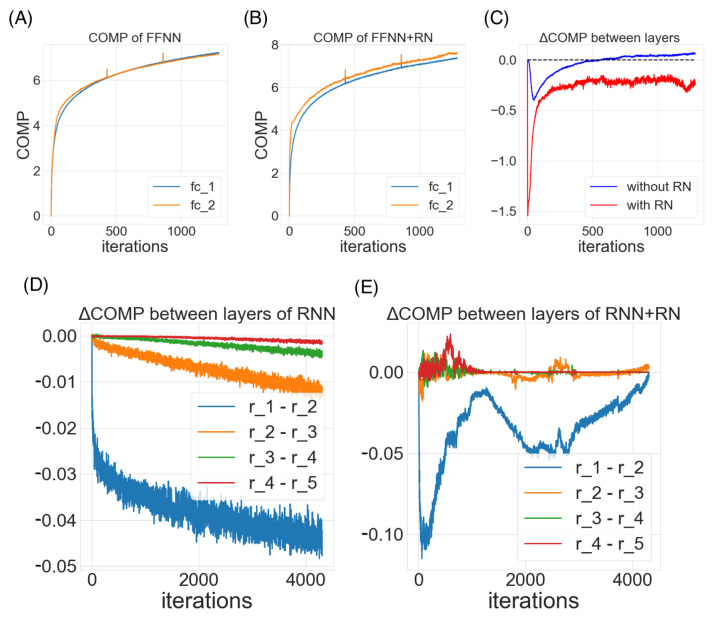
**Unsupervised attention mechanism as a probing tool:** In this demonstration, a classical 784-1000-1000-10 feedforward neural network (FFNN) and a vanilla recurrent neural network (RNN) with 100 hidden units over 5 time steps are trained for the MNIST classification task. The COMP of each layers (or unfolded layers in the recurrent time steps) are recorded throughout the training process. (**A**,**B**) are the COMP dynamics recorded for the fully-connected layers 1 and 2 in the feedforward networks without (**A**) or with (**B**) the regularity normalization. (**C**) The difference of COMP between the fully-connected layer 1 and 2 in the neural network with or without the regularity normalization. (**D**) The COMP difference between adjacent unfolded layers (time steps) in the recurrent neural network. (**E**) The COMP difference between adjacent unfolded layers (time steps) in the recurrent neural network with the regularity normalization.

**Figure 6 entropy-24-00059-f006:**
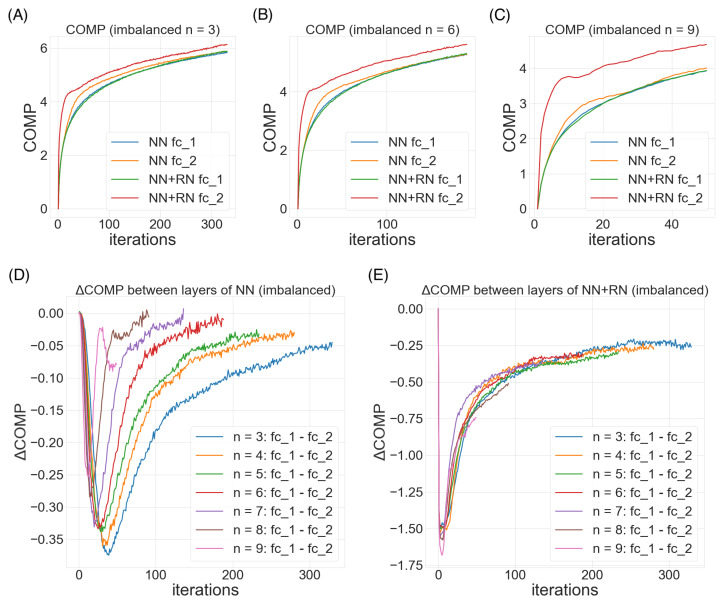
**Probing the unsupervised attention mechanism in imbalanced MNIST**: In the imbalanced MNIST task, we train a classical 784-1000-1000-10 feedforward neural network on the MNIST where *n* classes out of the ten are downsampled to 1%. We record the COMP the two fully connected (fc) layers of the neural network and compute the differences between the two (ΔCOMP). (**A**) The COMP dynamics of the neural network when 3 classes are downsampled. (**B**) The COMP dynamics of the neural network when 6 classes are downsampled. (**C**) The COMP dynamics of the neural network when 9 classes are downsampled. (**D**) The difference of COMP between the two layers in the neural net during training. (**E**) The difference of COMP between the two layers in the neural net during training when regularity normalization (RN) is applied.

**Figure 7 entropy-24-00059-f007:**
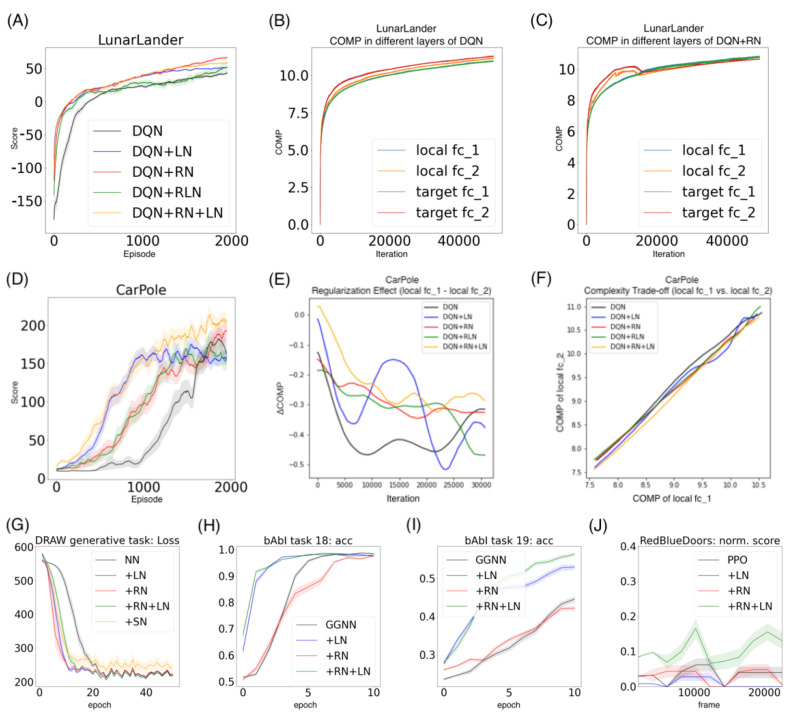
**Performance and analysis in additional tasks**: In the OpenAI gym environments, we train the Deep Q Networks with or without different types of normalization methods. (**A**) The learning curve of the score performance in LunarLander. (**B**) The COMP dynamics of different layers of DQN during the training in LunarLander. (**B**) The COMP dynamics of different layers of DQN with regularity normalization during the training in LunarLander. (**D**) The learning curve of the score performance in Carpole. (**E**) The regularization effect demonstrated by the COMP difference between the fully-connected layers 1 ($fc_{1}$) and 2 ($fc_{2}$) in the local Q networks. (**F**) The complexity trade-off demonstrated by the the COMP of $fc_{1}$ vs. $fc_{2}$. (**G**) The performance in the generative modeling of MNIST dataset with the Deep Recurrent Attentive Writer (DRAW) with or without different types of normalization methods, demonstrated by the KL divergence loss of across training epochs. (**H**,**I**) are the accuracy performance of the Graph Gated Neural Networks (GGNN) with or without different types of normalization methods in the task 18 and 19 in the bAbI benchmark, the most challenging two tasks. (**J**) The normalized score performance of the Proximal Policy Optimization (PPO) network with or without different types of normalization methods in the RedBlueDoors memory task, the most challenging task in MiniGrid benchmark.

**Table 1 entropy-24-00059-t001:** **Heavy-tailed scenarios:** test errors (in %) of the imbalanced MNIST 784-1000-1000-10 task.

	“Balanced”	“Rare Minority”			“Highly Imbalanced”				“Dominant Oligarchy”	
	n=0	n=1	n=2	n=3	n=4	n=5	n=6	n=7	n=8	n=9
baseline	4.80±0.15	14.48±0.28	23.74±0.28	32.80±0.22	42.01±0.45	51.99±0.32	60.86±0.19	70.81±0.40	80.67±0.36	90.12±0.25
BN	2.77±0.05	12.54±0.30	21.77±0.25	30.75±0.30	40.67±0.45	49.96±0.46	59.08±0.70	67.25±0.54	76.55±1.41	80.54±2.38
LN	3.09±0.11	8.78±0.84	14.22±0.65	20.62±1.46	26.87±0.97	34.23±2.08	36.87±0.64	41.73±2.74	41.20±1.13	41.26±1.30
WN	4.96±0.11	14.51±0.44	23.72±0.39	32.99±0.28	41.95±0.46	52.10±0.30	60.97±0.18	70.87±0.39	80.76±0.36	90.12±0.25
RN	4.91±0.39	8.61±0.86	14.61±0.58	19.49±0.45	23.35±1.22	33.84±1.69	41.47±1.91	60.46±2.88	81.96±0.59	90.11±0.24
RLN	5.01±0.29	9.47±1.21	12.32±0.56	22.17±0.94	23.76±1.56	32.23±1.66	43.06±3.56	57.30±6.33	88.36±1.77	89.55±0.32
LN+RN	4.59±0.29	8.41±1.16	12.46±0.87	17.25±1.47	25.65±1.91	28.71±1.97	33.14±2.49	36.08±2.09	44.54±1.74	82.29±4.44
SN	7.00±0.18	12.27±1.30	16.12±1.39	24.91±1.61	31.07±1.41	41.87±1.78	52.88±2.09	68.44±1.42	83.34±1.85	82.41±2.30

## Data Availability

The data and codes to reproduce all empirical results can be accessed at the GitHub repository https://github.com/doerlbh/UnsupervisedAttentionMechanism (accessed on 17 November 2021).

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
