# Peer review of "Regularity Normalization: Neuroscience-Inspired Unsupervised Attention across Neural Network Layers†"

_entropy, 2021, doi:10.3390/e24010059_

Round 1

Reviewer 1 Report

The authors propose the regularity normalization (RN) as an unsupervised attention mechanism (UAM) which computes the statistical regularity in the implicit space of neural networks under the Minimum Description Length (MDL) principle. Treating the neural network optimization process as a partially observable model selection problem, the regularity normalization constrains the implicit space by a normalization factor, the universal code length. We compute this universal code incrementally across neural network layers and demonstrate the flexibility to include data priors such as top-down attention and other oracle information. Experimental results show that the proposed method appears to be effective compared with related methods. Detailed comments are listed as follows,

The implementation of the proposed network is unclear. The authors should details these contents.

The definition of some notations are unclear, such Eq(2).

How to training the proposed mode? The optimization of the model should be given.

The authors claimed the proposed method is unsupervised. However, the working manner of the neuroscience in human brain is not unsupervised in most cases.

Adding some missing works related to this work

Some works also explore the attention mechanism in deep learning model, such as

Fei Wang, Mengqing Jiang, Chen Qian, Shuo Yang, Cheng Li, Honggang Zhang, Xiaogang Wang, and Xiaoou Tang. "Residual attention network for image classification." In Proceedings of the IEEE conference on computer vision and pattern recognition, pp. 3156-3164. 2017.

Jun Fu, Jing Liu, Haijie Tian, Yong Li, Yongjun Bao, Zhiwei Fang, and Hanqing Lu. "Dual attention network for scene segmentation." In Proceedings of the IEEE/CVF Conference on Computer Vision and Pattern Recognition, pp. 3146-3154. 2019.

Spatiotemporal co-attention recurrent neural networks for human-skeleton motion prediction. IEEE Transactions on Pattern Analysis and Machine Intelligence (TPAMI). 2021

Host-Parasite: Graph LSTM-In-LSTM for Group Activity Recognition. IEEE Transactions on Neural Networks and Learning Systems (TNNLS), 32(2): 663-674, 2021

Coherence Constrained Graph LSTM for Group Activity Recognition. IEEE Transactions on Pattern Analysis and Machine Intelligence (TPAMI), 2019

Author Response

Dear Reviewer,

Thank you for the careful read and constructive feedbacks. We really appreciate that you find our idea interesting and effective. We have revised our manuscript to reflect the suggestion that you kindly pointed out.

In the latest revision, we have improved our manuscript in the following ways:

Methodology: we have updated the problem setting and method sections to include more technical details regarding the mathematical formulations with more clarifications of the notations, code implementations with repository, and optimization and deployment settings with a flowchart.

Optimization: The proposed normalization approach is agnostic to the optimization method and feedback signals due to its unsupervised nature. Therefore, most reasonable learning methods for the neural net should be by default compatible to the proposed normalization method. The current optimization method used for the neural net training in our empirical evaluations is the stochastic gradient descent. We have now included a brief discussion in the method section to clarify this point, as well as a flowchart outlining where the role of the optimization within the framework. 

Unsupervised: We agree that the working manner of the neuroscience in human brain is not unsupervised in most cases, especially in the events of top-down control. The inspiration we primarily draw from the neuroscience, is the bottom-up neuronal firing patterns driven by the regularity. As a result, the proposed method consists of an unsupervised attention mechanism. And the discussion of whether the human brain is driven by supervised vs. unsupervised mechanism is not a focus of this work. Despite this distinction, we introduced the flexibility for the proposed methods to include supervised signals and feedbacks, as in the saliency normalization variant. We also updated the inspiration section to reflect this clarification. Thank you for the suggestion.

Related work: We have updated the related work section to include the five suggested references and discussed their relevance. Thank you for bringing these important references to our attention.

Thank you again for reviewing our manuscript and very helpful advice to help us improve our manuscript. We wish you a happy holiday season.

Reviewer 2 Report

The author proposes a variation of the MDL principle and they propose to extend this approach for feedforward neural networks. 

The introduction is extensive and well-written. And despite I personally find that readers may get lost by so much information provided, I find it interesting and relatively well-motivated.

This approach provides an interesting point on tackling imbalanced data, and if it is replicable it can be useful for further model development.

It seems like the authors spend a lot of time trying to justify many key points in the paper, and so, I would recommend another read prior to publication by the author to find minimal typos and grammar errors. If authors find it necessary, some long sentences could be rewritten to make it easy for the reader to understand what the author wants to tell. It can be tough to read the paper, but the information provided seems relevant. 

I like that author provided a link to the code.

For me, this paper could be published after a final reread.

Author Response

Dear Reviewer, 

Thank you for the careful read and positive feedbacks. We really appreciate that you find our idea novel and interesting, our introduction well-motivated, and the discussed concepts relevant. 

In the latest revision, we have reread our manuscript to fix some minimal typos, shorten certain long sentences to facilitate the reading, as well as introduced a few more technical details as requested by other reviewers.

Thank you again for reviewing our manuscript and very helpful advice to help us improve our manuscript. We wish you a happy holiday season.

Reviewer 3 Report

The paper is interesting and it is technically sound.

However this manuscript needs to be better organized: pseudocode and flochart(s) should be provided for the proposed methodology.

The Author needs to add a Conclusion section (without references).

Finally, progress with respect to the state of the art should be better shown.

Author Response

Dear Reviewer, 

Thank you for the careful read and constructive feedbacks. We really appreciate that you find our idea interesting and technically sound. We have revised our manuscript to reflect the suggestion that you kindly pointed out. Please see below for the answers to your questions.

Method description: We have revised our method section to include more details on various technical points. The new version also includes the opensourced code base and a flowchart for the training and deployment of the proposed approaches to facilitate the understanding and reproducing of the proposed method.

The Conclusion section (without references): We have now reorganized our manuscript to include a dedicated conclusion section without references. 

Progress with respect to the state of the art: We have updated the related work section to include a more extensive discussion of several new references of the state-of-the-art attention-based approaches and their relevance to our work. 

Thank you again for reviewing our manuscript and very helpful advice to help us improve our manuscript. We wish you a happy holiday season.

Reviewer 4 Report

This manuscript focuses on the regularity normalization for neuroscience-inspired unsupervised attention across neural network layers. There are some issues in it.

1.The regularity layer normalization (RLN) is proposed in section 2.2, but the regularity normalization (RN) is proposed in the title. Which one is correct?
2.The probability in line 126 is P(x) and the probability in line 140 is p(x). Are they the same?
3.There is no log(exp) in Eqs. (1) and (2), but there is log(exp(P)) in line 4 in Algorithm 1.
4.Eq.(1) and Eq.(3) seem to be inconsistent. The summation in Eq. (1) is for all x', but the summation in Eq. (3) is for j<=i.
5.The parameters x and "theta" above Eq. (3) have a superscript and a subscript, but the The parameters x and "theta" in Eq. (3) only have a subscript.
6.The parameters x and "theta" in Eq. (3) have the same subscript "j", but the parameters x and "theta" in line 192 have different subscripts "i" and "t".
7.All the "theta"s in lines 149, 158, 162, and 189 need "^".
8.What is the unit for the data in Table 1?
9.According to Table 1, RN performs best only for the case with n=4. Why is RN emphasized in the title?
10.According to Table 1, it is difficult to say which one can perform well always. Whereas, LN+RN seems to perform better than RN and RLN proposed by this manuscript  for “Highly imbalanced” cases.

In conclusion, major revision is needed.

Author Response

Dear Reviewer, 

Thank you for the careful read and constructive feedbacks. We have revised our manuscript to reflect the suggestion that you kindly pointed out. Please see below for the answers to your questions.

1.The regularity layer normalization (RLN) is proposed in section 2.2, but the regularity normalization (RN) is proposed in the title. Which one is correct?

They are both our proposed methods within a family of regularity-based normalization methods. As outlined in more details in our method section (section 5), the regularity normalization (RN) is the standard formulation that we proposed (section 5.1). The regularity layer normalization (RLN), along with regularity batch normalization (RBN) and saliency normalization (SN), are the variants of this class of regularity-based normalizations, detailed in section 5.2 and 5.3. We agree that the reference to the RLN in our related work section (section 2.2) can potentially be confusing, so we have now further clarified this confusion by also directly referencing our standard method regularity normalization in section 2.2. Thanks for pointing this out.

2.The probability in line 126 is P(x) and the probability in line 140 is p(x). Are they the same?

The P(x|\theta) in line 126 is not the same as the p(x) in line 140. The P(x|\theta) in line 126 is the probability of observing the data sample x given the model parameter \theta, hence the capital P. The p(x) in line 140 is a generic distribution notation that is used to describe the minimax regret formulation. The standard usage of notations f, p and q are the same as used in the reference [28] Myung et al., 2006.

3.There is no log(exp) in Eqs. (1) and (2), but there is log(exp(P)) in line 4 in Algorithm 1.

Yes, that is correct. As introduced in section 5.1, the log(exp), or the LogSumExp (LSE) or RealSoftMax, is a standard numerical trick that is used to compute the log sum of two values in a stable way. We have now provided more context and reference to this stablization technique in section 5.1. 

4.Eq.(1) and Eq.(3) seem to be inconsistent. The summation in Eq. (1) is for all x', but the summation in Eq. (3) is for j<=i.

Thanks for the observation. The difference between the Eq. 1 and Eq. 3 is intentional. In the standard formulation of P_NML in Eq. 1, the minimax solution of a certain data sample x is obtained by normalizing the probabilty of that data sample x given the optimal model parameter \hat{\theta(x)} against the sum of that of all possible data sample x'. This is a different setting in our empirical case of neural network optimization problem. In Eq. 3, since the neural net is optimizing in an incremental way, the available data samples used to compute the normalization denominator are only the data samples that have been seen so far by the model at optimization step i, thus it is j from 0->i. We have now provided more contexts and descriptions around both Eq. 1 and Eq. 3 to clarify on this point.

5.The parameters x and "theta" above Eq. (3) have a superscript and a subscript, but the The parameters x and "theta" in Eq. (3) only have a subscript.

Thank you for the careful read. The notations of x and \theta that have both the superscript and subscript are the ones used in Figure 3. The superscript are used to refer to the network layer or computing module index of the neural net (e.g. x^3_1 would mean the data sample seen by the layer 3 of the neural net in optimization step 1). In Eq. 3 on the other hand, is a more generic formulation of any neural network modules that has no reference to the layer or module index. Thus, the notations of x and \theta only has a subscript. We agree that leaving the superscript out in later formulation would cause unnecessary confusion, so we have now added more explicit description right above Eq. 3 to clarify this point. Thank you for pointing it out.

6.The parameters x and "theta" in Eq. (3) have the same subscript "j", but the parameters x and "theta" in line 192 have different subscripts "i" and "t".

We agree that reusing the notation i and j in different contexts could cause unnecessary confusion. We keep the t notation below \theta and COMP to facilitate the readers' understanding of the neural net's state, which may or may not be the same as the training batches. Other than this, we have updated all other notations including the legend in Figure 3 and descriptions in section 4 to reflect a more consistent notation as suggested. Thank you for discovering this important point.

7.All the "theta"s in lines 149, 158, 162, and 189 need "^".

Thank you for the very careful read and finding this typo. We have now corrected the \theta in lines 158, 162, and 189 to be \hat{\theta} as suggested. The \theta in line 149 is intentionally hat-less: in line 149, we talked about \theta as a generic model parameter within the model class \Theta, that are not necessarily optimal or estimated.

8.What is the unit for the data in Table 1?

The unit is the percentage. We have now included this important details in our legend. Thank you.

9.According to Table 1, RN performs best only for the case with n=4. Why is RN emphasized in the title?

We emphasize the regularity normalization because we consider the regularity normalization as a class of methods that utilize the regularity to reparameterize the models. The RN, RLN, and SN etc. are variants of this method family, with the RN as its standard case.

10.According to Table 1, it is difficult to say which one can perform well always. Whereas, LN+RN seems to perform better than RN and RLN proposed by this manuscript  for “Highly imbalanced” cases.

Thank you and that's a very good observation, because it is exactly one of the questions that we are investigating with this imbalanced setting: how do different normalizations affect the learning at different levels of imbalanceness. Yes, in certain cases, like n=0, 8 and 9 ("balanced" or "dominant oligarchy"), some of the baselines are doing better than RN. However, that is expected because those are the three cases in the ten scenarios that the dataset is more "regular" (or homogenous in regularity), and the benefit from normalizing against the regularity is expected to be minimal. On the other hand, if we look at the test accuracy results in the other seven scenarios, especially in the highly imbalanced scenarios (RN variants over BN/WN/baseline for around 20%), they should provide promises in the proposed approach ability to learn from the extreme regularities. 

This also brings up one interesting concept in the applied statistics, the "no free lunch" (NFL) theorem. LayerNorm, BatchNorm, WeightNorm and the proposed Regularity Normalizations are all developed under different inductive bias that the researchers impose, and these different inductive bias would yield different performance in different scenarios. In our case, the assumption that we make, when proposing the regularity-based approach, is that in real life, the distribution of the encounter of the data samples are not necessarily i.i.d, or uniformly distributed, but instead follows irregularities unknown to the learner. The data distribution can be imbalanced in certain learning episodes, and the reward feedback can be sparse and irregular. This inductive bias motivates the development of this regularity-based method. 

In addition, we believe the main point is not simply about beating the state-of-the-art normalization method with another normalization, but more to offer a new perspective where people can gain insights of the deep network in action – through the lens of model complexity characterized by this normalization factor the model computes along the way. On the subsidiary numerical advantage of the proposed method, the results suggest that combining regularity normalizations with traditional normalization methods can be much stronger than any method by itself, as their regularization priors are in different dimensions and subspaces. We have reworded our language to better reflect this statement in both the introduction as well as the result sections.

Thank you again for reviewing our manuscript and very helpful advice to help us improve our manuscript. We wish you a happy holiday season.

Round 2

Reviewer 3 Report

The Authors have addressed all of my comments and suggestions in this round of review. I recommend publishing this manuscript as it is.

Reviewer 4 Report

The revised manuscript is ready to be published.